# Virus-Based Biological Systems as Next-Generation Carriers for the Therapy of Central Nervous System Diseases

**DOI:** 10.3390/pharmaceutics15071931

**Published:** 2023-07-11

**Authors:** Ilona Nowak, Marcel Madej, Julia Secemska, Robert Sarna, Barbara Strzalka-Mrozik

**Affiliations:** Department of Molecular Biology, Faculty of Pharmaceutical Sciences in Sosnowiec, Medical University of Silesia, 40-055 Katowice, Poland; mc.ilona.nowak@gmail.com (I.N.); mmarcel281297@gmail.com (M.M.); j.secemska@gmail.com (J.S.); robert90sarna@gmail.com (R.S.)

**Keywords:** viral vector, drug delivery system, central nervous system, Alzheimer’s disease, Parkinson’s disease, glioblastoma multiforme, multiple sclerosis, Canavan disease, virus-like particles, blood–brain barrier

## Abstract

Central nervous system (CNS) diseases are currently a major challenge in medicine. One reason is the presence of the blood–brain barrier, which is a significant limitation for currently used medicinal substances that are characterized by a high molecular weight and a short half-life. Despite the application of nanotechnology, there is still the problem of targeting and the occurrence of systemic toxicity. Viral vectors and virus-like particles (VLPs) may provide a promising solution to these challenges. Their small size, biocompatibility, ability to carry medicinal substances, and specific targeting of neural cells make them useful in research when formulating a new generation of biological carriers. Additionally, the possibility of genetic modification has the potential for gene therapy. Among the most promising viral vectors are adeno-associated viruses, adenoviruses, and retroviruses. This is due to their natural tropism to neural cells, as well as the possibility of genetic and surface modification. Moreover, VLPs that are devoid of infectious genetic material in favor of increasing capacity are also leading the way for research on new drug delivery systems. The aim of this study is to review the most recent reports on the use of viral vectors and VLPs in the treatment of selected CNS diseases.

## 1. Introduction

Central nervous system (CNS) diseases, including Alzheimer’s disease (AD), Parkinson’s disease (PD), Canavan disease (CD), multiple sclerosis (MS) and those associated with the neoplastic process in astrocytic series cells, such as glioblastoma multiforme (GBM), are a major problem in modern medicine [1,2]. The reason for this is that the current medicinal substances used for treatment show relatively low efficacy due to their high molecular weight, poor ability to target a specific cell type and difficulty crossing the blood–brain barrier (BBB) [3,4,5]. The problem of permeability across the biological barrier is a result of several features. The first is the existence of highly functional tight junctions between CNS endothelial cells, which prevent free paracellular passage from the circulatory system to the brain [6,7,8,9]. Since the BBB is highly selective, its additional feature is the presence of transporters on the surface of CNS endothelial cells [8,10]. In addition, CNS endothelial cells exhibit an extremely low rate of transcytosis, which contributes significantly to limiting transport through the vessel wall [10]. Nevertheless, some macromolecules, such as insulin, albumin, low-density lipoproteins, and iron-bound transferrin, have the ability to penetrate the brain via transcytosis [11]. This can be either receptor mediated or adsorption mediated [11,12].

The effectiveness of the delivery of medicinal substances to the CNS therefore depends on the size of the molecule, which should be below 200 nm, and the lipophilicity, polarity, and charge for optimal transport across the BBB [13]. Lipophilic molecules show better BBB penetration capabilities, while hydrophilic drugs require specific transport strategies [14,15,16,17]. Currently used drugs often have difficulty maintaining the aforementioned properties, contributing to the low efficiency of therapy [18]. To transport drugs to the CNS, strategies based on nanotechnology solutions are currently being utilized, using the incorporation of medicinal substances into nanoparticles based on polymeric materials [18]. An increasing number of reports also suggest a high potential for the use of viral vectors as delivery systems for medicinal substances in CNS-related diseases [19,20].

According to the literature, viral vectors and virus-like particles (VLP) may offer hope for new methods of delivering medicinal substances to treat CNS diseases [19,20]. Their small size (which allows them to cross the BBB easily), ability to infect neural cells, biocompatibility, and ease of viral manipulation mean that they are increasingly being used in the study of drug delivery systems (DDS) for CNS diseases [19,20]. Among them, lentiviruses, adenoviruses, retroviruses and adeno-associated viruses (AAVs) have led the way. Studies have been conducted with them on the effective delivery of drugs, as well as on the delivery of small molecules such as microRNAs (miRNAs) or small interfering RNAs (siRNAs) to CNS cells in both cellular and animal models [21]. In addition, the ability to modify the surface of viral capsid proteins creates even more opportunities for cellular targeting and prolongs their presence in the body [21]. Interestingly, due to their lytic cycle, some viruses may themselves provide a therapeutic solution. Another promising prospect is the formulation of virus-like particles without infectious genetic material in favor of increasing accessibility to the incorporated medicinal substance to be delivered to a specific site in the body [19].

With regard to this information, the purpose of this review is to present the latest developments, advantages, limitations and prospects for the use of virus-based drug delivery systems in the treatment of selected CNS diseases.

## 2. Limitations of Currently Used Drug Delivery Systems to the Central Nervous System

### 2.1. Conventional Drug Delivery Systems

Drug delivery to the brain and spinal cord is hindered by the physiological BBB. Therefore, combinations of a medicinal substance (e.g., methotrexate, cytarabine, or carmustine) with osmotically enhancing agents that increase the permeability of the BBB, e.g., mannitol, are currently being used, leading to the better penetration of the drug through the vascular endothelium [22,23]. Typically, in drug delivery systems, medications are administered orally, intranasally, rectally, mucosally, or by injection. Unfortunately, such administered drugs are less well absorbed, and are distributed randomly throughout the body and damaged areas not affected by the disease; they require longer administration times to cure the disease. Moreover, they are less effective due to rapid enzymatic degradation or differences in pH [22,24]. One solution to this problem is convection-enhanced delivery (CED) mass fluid flow, which depends on the external pressure gradient provided by the syringe pump [22,24]. Thus, it ensures the continuous infusion of drug formulations, such as drug-containing nanoparticles, to the site of action. This solution significantly increases the accumulation of a large volume of therapeutic agents in the brain, resulting in improved therapeutic effects [25]. Another promising concept is implantable and biodegradable rods based on poly(l-lactide-*co*-glycolide-*co*-trimethylene carbonate) (P(l-LA:GA:TMC)), with aripiprazole (ARP) administered via a ampoule syringe as an alternative to small microparticles administered as suspensions [26]. It is also worth mentioning the use of terpolymer-shaped memory rods, which, in addition to their use in CNS diseases, can be used in nerve regeneration [27].

Drug delivery systems can be divided by the mode of administration: invasive and noninvasive. Invasive approaches include the modulation or interruption of the BBB, intracerebral implants, and intracerebral injections or infusions [22]. However, these methods cause discomfort to patients and are not preferred, as other less severe treatments are available. Noninvasive methods include the use of medications taken orally or intranasally [22].

### 2.2. Nanoformulations

In the past decade, research has confirmed that nanoformulations are being used successfully in the treatment of CNS diseases due to their small size (1 to 1000 nm), their increased stability, extended circulation time in the blood by polyethylene glycol coating, controlled drug release, and targeting effects [25]. Therefore, depending on the application, nanomaterials can be used as drug carriers, nerve growth stimulators, diagnostic tools, and used to modulate interactions between neural cells [25].

Nowadays, the most successful nanoformulations are those with a prolonged drug release, which are mainly made of fully biodegradable polymeric materials. One example is the form of the drug incorporated into nanoparticles made of poly(d,l-lactide-*co*-glycolide) (d,l-PLGA), poly(ε-caprolactone), or poly(d,l-lactide) [25].

The most commonly used forms in the treatment of brain-related diseases are nanoparticles, which are capable of crossing the BBB due to characteristic properties that are selected appropriately during their formulation [13]. Relatively important in their formulation are properties such as size, shape, molecular weight, polymer molar ratio or zeta potential, and stability in a biological system. Their small size allows the drug carrier to penetrate blood vessels or target tissues, making it easier to reach specific sites in the body [13,24]. The molar ratio of the polymer, in turn, affects the physical and chemical properties of the material, determining the ability of the carrier to capture, store, and control drug release [13]. The zeta potential plays a key role in carrier stabilization, controlled drug release, interactions with cell membranes, and physicochemical stability in the context of a biological system [13,18]. Among the nanoformulations can polymer nanoparticles, mesoporous nanoparticles, carbon nanotubes, dendrimers, liposomes, and metallic nanoparticles be distinguished. Liposomes were among the first drug carriers to be explored, including nano/micromolecular or colloidal spherical vesicles ranging in size from 80 to 300 nm. They consist of phospholipids and steroids (e.g., cholesterol), bilayers, or other surfactants [28]. Dendrimers, on the other hand, as unique polymers, are characterized by a regular, branched structure that allows the modification of many functional groups compared to linear polymers. They consist of a core, dendrons, and surfactant groups, which determine the biocompatibility and physicochemical properties of dendrimers [29]. Carbon nanotubes, like dendrimers, have a huge surface area obtained by rolling single or multiple layers of graphite. In addition, they have excellent electronic and thermal conductivity [30].

All the aforementioned structures have unique properties that find applications in various fields. One example of the use of nanoparticles in the treatment of CNS diseases is the delivery of drugs into the brain to treat AD [13]. In this case, nanoparticles with an incorporated medicinal substance (e.g., doxorubicin, paclitaxel, or cisplatin) in PLGA are leading the way [13,31]. Additionally, nanoparticles can be used as carriers for drugs such as acetylcholinesterase inhibitors or anti-amyloid antibodies, which, when delivered to the brain, prevent the accumulation of beta amyloid deposits, thereby improving cognitive function [31,32].

In addition to nanoparticles, approaches using proteins are increasingly being used. For example, shuttle peptides, which are discovered using phage presentation technology or are derived from natural neurotrophic proteins and some viruses, have also shown promise in overcoming the BBB. Brain-permeable peptide–drug conjugates (e.g., doxorubicin, or paclitaxel conjugated with angiopep 2 peptide), comprising BBB shuttle peptides, linkers, and drug molecules, are becoming a promising system for delivering drugs to the CNS across the BBB barrier [32].

### 2.3. Gene Therapy and Drug Delivery Systems

In the treatment of numerous diseases, not only those related to the nervous system, the potential of gene therapy, which aims to treat the disease itself and not only its symptoms, is increasingly being exploited [33]. In such therapy, it is important to construct suitable vectors that translocate across biological membranes and protect genes from the degenerative effects of the environment [33]. Vectors can be divided into two types: viral and nonviral vectors. This classification is shown in Figure 1.

The selection of a vector depends on the type and amount of genetic material, site, and route of administration. Each has its own individual advantages and disadvantages. Nonviral vectors, which we can divide by the method of obtainment (physical, chemical), are less toxic and are capable of transferring large amounts of genetic material, but have great difficulty overcoming extracellular and intracellular barriers, a reduced transfection capacity, and considerably lower transgene expression [33]. However, viral carriers (adenoviral vectors, retroviruses, lentiviruses, adeno-associated viruses, and poxviruses) provide good transfection efficiency and sustained gene expression, as well as the excellent protection of the gene from degradation [33]. This makes them the most promising approach in CNS diseases as next-generation carriers [33]. In this case, the gene is directed to the affected site, such as cells in the eye or brain, or directly into the blood. Such a method can allow the delivery of not only drugs, but also naked DNA and various types of RNA, such as siRNA or miRNA [33]. However, the main disadvantage of this method is the low gene expression of naked genetic material [33].

## 3. Advantages of Virus-Based Biological Systems as Next-Generation Carriers for the Therapy of Central Nervous System Diseases

### 3.1. Viruses Used as Drug Delivery Systems for the Central Nervous System

Viral vectors are being used in the development of new drug delivery strategies due to their ability to precisely target and penetrate their well-defined cellular targets [21]. Viruses also possess the ability to easily express transmitted genetic information in host cells [21]. The state-of-the-art strategy for generating viral vectors is to replace wild-type viral genes that determine lysis and cytotoxicity with therapeutic genes. Cell lines and helper viruses are used to generate vectors with the transgene [21]. The most commonly used viruses as drug delivery systems for CNS are retroviruses (RVs), e.g., *human immunodeficiency virus* (HIV), adenoviruses (AVs), lentiviruses (LVs), and AAVs [21]. The aforementioned viruses vary in their payload, cell tropism, and capacity for and their persistence of transgene induction [21]. In the case of viruses, cell tropism (i.e., preferential infection of specific cells in the organism) is essential for specific drug delivery [21]. It differs due to the presence of virus-type-specific proteins, through which it is able to bind to receptors present on the surface of the target cell [21]. The type of tissue infected also largely depends on the serotype of the virus [21]. It is assumed that AV preferentially infects neuronal cells, dendritic cells, and hepatocytes. A similar tropism is shown by AAVs, which also infect myocytes. RV, on the other hand, targets proliferating cells and quiescent cells [21].

Adenoviruses have a genome size of 35–40 kb (30–40 genes) [21]. The delivered linear DNA can be no larger than 7.5–8 kb [34]. This type of virus effectively infects dividing cells and does not integrate into the host genome [21,34]. However, the most significant limitations of these vectors include their high immunogenicity [21,34], gene expression of 2 weeks to several months (transient), and high risk of cytopathic effects [21,34]. Additionally, high titers can result in organ damage and even mortality [34]. However, these disadvantages are easily corrected by designing and synthesizing novel serotypes [21]. AV has found applications in the treatment of cancer, Parkinson’s disease, and Huntington’s disease [21]. Huang et al. [35] used an oncolytic adenoviral vector carrying IL-7 (oAD-IL7) in combination therapy with B7H3-targeted chimeric antigen receptor T-cell (CAR-T). Both therapies, separately and in combination, were tested on in vivo and in vitro models using glioma cell lines, including U87, U251, A172, and T98G, and the engineered GBM19-LUCF line [35]. Six- to eight-week-old xenografted triple immunodeficient mice (NCG mice) were also used in the study. The study showed that the applied adenovirus improved the effects of CAR-T therapy in both in vitro and in vivo conditions; however, independently in the in vitro model, it showed the effective promotion of the apoptosis of glioma cell lines [35].

LVs, compared to AVs, are capable of infecting both dividing and nondividing cells while preserving long-term transgene induction and maintaining a safe immune profile [21]. However, they easily undergo insertional mutagenesis, although splitting the viral genome into individual plasmids is used to minimize the risk of this occurring [21,34]. Genetic engineering makes it possible to produce LVs with specific integration sites for the increased safety of use. They are being used for the treatment of PD and AD [21]. Chen et al. [36] applied an LV vector carrying the transcription factors *Ascl1*, *Bm2*, and *Ngn2* (ABN) (LV-ABN) via intrahippocampal injection using a mouse model of AD (adult male C57BL6 mice), administering a 2 µL dose of viral suspension with a titer of 1.5 × 10^9^ pfu/mL. The results showed a decrease in spatial learning and memory deficits in infected mice. Therefore, this virus may have the potential to alleviate learning and memory impairment [36].

AAV vectors are small, nontargeted units with genome DNA payloads < 5 kb [21,34]. Of the viral vectors, they are the most promising due to their ability to transduce dividing and nondividing cells [21,34]. Transduction by these vectors demonstrates high clinical safety while maintaining a long transgene expression [21,34]. AAVs have an uncomplicated three-gene genome (*rep*, *aap*, and *cap* genes) [21]. These genes determine viral replication, integration, and packaging [21]. The transduction abilities, tropism, or the ability of AAVs to cross the BBB depend on serotypes exhibiting different capsid protein compositions [21]. Serotypes can be wild-type or recombinant via the capsid and genome mixing techniques of different serotypes, the insertion of peptide motifs from phage libraries, and the random incorporation of peptide motifs [21]. These vectors are versatile and enable one to test for many CNS diseases [21]. GuhaSarkar et al. [37] treated nude thymic male mice with the AAV9/CB-hiFNβ vector to test the effect of FNβ on glioma cells. The vector was administered systemically by tail vein injection at a dose of 200 µL and intracranially via the infusion of 2 µL with a titer of 7.6 × 10^9^ pfu/mL [37]. This study showed that the systemic administration of AAV9/CB-hiFNβ can inhibit tumor growth and effectively treat invasive glioblastoma multiforme tumors. Therefore, it seems promising to use such a delivery system for the treatment of GBM [37].

Some examples of the use of these viruses for the treatment of CNS diseases are shown in Table 1.

### 3.2. Methods of Viral Vector Formulation

Success in therapy depends largely on the carriers; therefore, it is crucial to properly design and fabricate viral nanocarriers to deliver drug substances [45]. Some naturally occurring viral variants, due to their innate properties, have already been clinically tested [46]. Improving the characteristics of viral vectors involves two main directions: capsid engineering and the genetic engineering of the viral genome. Additionally, altered viruses can contain both natural (found in viruses or other organisms) and synthetic (such as polymers or inorganic nanoparticles) components (Figure 2) [46].

Depending on the therapeutic application, the desired functions of vectors may be limited to targeting altered cells or the co-delivery of small-molecule drugs (e.g., donepezil, temozolomide, or perampanel) [45,46]. Molecules such as biotin or small-molecule drugs can be attached to the viral capsid and act as adaptors or medicines. Wei et al. [47] linked paclitaxel, a chemotherapeutic drug, to the AAV capsid via a lysine present on its surface. This treatment did not adversely affect the efficiency of gene delivery by the AAV. Unfortunately, the combination of the virus with paclitaxel did not effectively destroy cancer cells [47].

A viral system that has proven effective in drug delivery is the use of bacteriophage MS2 [48]. For this type of virus, surface manipulation is used, where the most common strategy is the attachment of amino acids with characteristic side chains. Moreover, it is possible to encapsulate hydrophobic drugs (e.g., liposomal amphotericin B and paclitaxel) through phage self-organization and block copolymerization [49].

Synthetic viruses can be created by mixing existing parts of the virus, resulting in chimeric or mosaic capsids [46]. Chimeric viruses are created by the genetic fusion of the capsid genes of two or more viruses, resulting in a new virus with a hybrid capsid, while the mosaic method appears to be an effective method for combining the phenotypic characteristics of two or more serotypes [46,50]. Pseudotyping is another acceptable strategy for combining viral parts derived from different types of viruses. Most often, such fusion involves the modification of enveloped viruses (e.g., lentiviruses), although nonenveloped viruses can also be pseudotyped [46,51]. With this method, it is possible to alter the tropism of a virus by changing one viral envelope protein to another. In addition to the strategies of rational design described above, parts of the virus can be mixed in combination [46].

In DNA shuffling, chimeric virus assemblies are constructed by randomly fragmenting, assembling, and amplifying capsid genes from different viral variants [46]. Synthetic parts, such as polymers (e.g., polyethylene glycol) or inorganic particles (e.g., silver, gold, or carbon nanotubes), can be attached to viruses to give them new functions [46]. Genetically encoded peptides and proteins can be added to viruses at appropriate sites or randomly throughout the capsid to give them new functions. In contrast, the level of virulence can be altered by introducing point mutations in the capsid [46,50,51].

Most clinical protocols use recombinant RVs and AVs as gene carriers. In the production of nonreplicating first-generation adenoviral vectors, the inverted terminal repeat (ITR) sequence containing the replication start of Ad5 (ORIGIN), the packaging signal (the “ψ” sequence), the *E1* gene (first-generation vectors), and often the *E3* gene are removed [52]. In second- or higher generation vectors, more genes are deleted from the wild-type virus genome [52]. Deletions of the E1 region prevent the activation of the *E* and *L* genes, which significantly reduces transcription and initially inhibits viral DNA replication (Figure 3) [52].

LV vectors are produced by the calcium cotransfection of the HEK293T cell line with three plasmids: one containing the genome of the produced vector, one containing element-encoding capsid proteins, and one containing genes and regulatory elements that are important for cell proliferation [53]. The obtained and propagated lentiviral vectors are purified and concentrated on commercially available columns or by ultracentrifugation [53].

### 3.3. Mechanism of Gene and Drug Delivery Using Virus-Based Nanosystems

The mechanism involved in the delivery of the medical substance or molecules, i.e., proteins and nucleic acids, largely depends on the virus-based system used. There are minor variations in the use of specific viruses or virus-like particles [19,33]. These differences mainly include the process of internalization into the cell using different membrane receptors specific to the virus, the location of release in the cytoplasm or cell nucleus, and the process of integration into the host genome [19,33,54,55]. However, the general scheme follows most of the same steps. The first step involves the introduction of a therapeutic transgene or molecule into a viral vector, followed by its delivery, usually by injection, into cells. In the second stage, the vector internalizes with the cell by interacting with receptors present on the cell membrane surface of the target cells [19,54,55]. Transport across the membrane occurs with the involvement of the endosome, where the packed viral vector is then transported into the cytoplasm, and the capsid is then released from the endosomal vesicle [19]. Then, binding to nuclear pores, it passes into the cell nucleus. The transferred transgene undergoes integration into the host genome, followed by a process of transcription and translation, leading to the synthesis of a new therapeutic protein [19]. The integration of genetic material is possible only in the case of some viruses, i.e., LVs or RVs [19,33]. For adenoviruses, the process involves only the delivery of the therapeutic substance to the cytoplasm or cell nucleus, where the transgene undergoes transcription directly, independent of the host genome [19,33,54].

In addition, the delivery of genetic material by means of viral-based molecules provides considerable opportunities, since they are able to escape lysosomal degradation, which reduces the bioavailability of the transferred compound [19]. The mechanism of medicinal substance and gene delivery using virus-based biological carriers is shown in Figure 4.

## 4. Viruses as Drug Delivery Systems in Selected Central Nervous System Diseases

### 4.1. Alzheimer’s Disease

AD is a slowly progressive neurodegenerative disease that is a major cause of dementia. It involves the progressive loss of cognitive function, which includes memory loss and difficulty thinking, speaking, and solving logical problems [56]. In addition, the presence of senile plaques formed by extracellular deposits of beta-amyloid protein and neurofibrillary tangles, which are abnormal fibers of tau protein that have become hyperphosphorylated, affect neuronal transmission in the brains of patients. Beta-amyloid protein plays an important role in neurotoxicity and affects neuronal function, so large amounts can lead to axonal damage [57,58]. Only two drug therapy options are available for AD patients. The first is cholinesterase inhibitors, which are used to treat patients with dementia associated with AD [59]. In patients with moderate to severe disease, memantine, which exhibits the activity of both a noncompetitive N-methyl-D-aspartate receptor agonist and a dopamine agonist, has been approved for use [59].

Although the causes of most AD cases are not fully understood, the search for alternatives to classical pharmacotherapy has begun. Interleukin-2, which acts as a regulator of inflammation, was introduced into the AAV vector and used experimentally to alleviate AD symptoms in a mouse model of the disease [60]. The AAV–interleukin-2 complex was found to have the ability to remodel the hippocampus and improve synaptic conduction, resulting in a gradual recovery of memory capacity with rare side effects [60]. A study was also conducted with a reactive microglia cell line using AAV-mediated gene therapy to downregulate glial maturation factor gene expression. The inhibition of neuroinflammation progression after genome editing was observed, indicating that this type of AAV-mediated therapy may be a potential treatment for AD [60,61]. The overexpression or accumulation of amyloids by inducing inflammation leads to brain cell death, so the amyloid precursor protein is another potential target for AD treatment. CD74 has been shown to have the ability to bind to amyloid precursor proteins and inhibit beta-amyloid formation. The AAV-mediated expression of CD74 led to a decrease in beta-amyloid formation. A study was conducted on a mouse animal model, and the therapy used improved the animal’s brain function [60,62]. Previous studies have confirmed that AAVs are a safe and promising alternative treatment for AD.

### 4.2. Parkinson’s Disease

PD is an age-related neurodegenerative disease. Several specific motor symptoms are caused by the extensive and progressive loss of dopamine-containing neurons located in the black matter of the midbrain [63]. Currently, pharmacological treatment comprises replacing dopamine via the oral use of a dopamine precursor (levodopa in combination with carbidopa) or dopamine agonists, which prolong the activity of endogenous dopamine [64]. As the disease progresses, the drugs become less effective and very often cause additional side effects. An alternative to traditional treatment is the widely studied AAV therapy. The goal of treatment is to rescue damaged neurons by delivering appropriate neurotrophic factors or to restore nerve cell function by delivering the essential enzymes responsible for dopamine synthesis and metabolism [60].

Neurturin belongs to the family of neurotrophic factors of glial origin. As an endogenous trophic factor, it significantly increases dopamine activity in the midbrain, improves neuronal survival, and protects the integrity of neurons from neurotoxic damage, thus potentially inhibiting the progression of PD [65]. CERE-120 is an experimental drug comprising an AAV combined with the neurturin gene. During the clinical trial phase, the agent was administered directly into the black matter of the midbrain [60].

Another promising treatment is the use of aromatic l-amino acid decarboxylase (AADC), an inhibitor of the premature conversion of levodopa to dopamine. A phase I clinical trial using adeno-associated virus serotype 2 delivering AADC (AAV2-AADC) has shown that such gene therapy is safe [60]. This treatment modality, in combination with other mechanisms, may be a promising future target for research on the treatment of PD [60,66].

Several clinical trials using viral vectors in the treatment of PD and AD are presented in Table 2, generated from the https://clinicaltrials.gov database (accessed on 3 April 2023).

### 4.3. Multiple Sclerosis

MS is a multifocal neurodegenerative disease of the central nervous system that has an autoimmune basis. The immune system attacks the myelin sheath, leading to inflammation and nerve damage. It often leads to physical disability, cognitive impairment, and a reduced quality of life [67]. The treatment of MS can be divided into three categories: treatment of acute relapses, disease-modifying therapies (DMTs), and symptomatic treatment [68].

Recurrent relapses of MS require immunoregulatory therapy. In one study, blocking the CD28-B7 and CD40-CD40L costimulatory pathways was shown to be an effective method for alleviating experimental autoimmune encephalomyelitis (EAE) in a mouse model of MS [69,70]. For this purpose, AAV was used as a carrier vector to deliver CTLA1 immunoglobulin (Ig) or CD40-Ig to EAE induced by myelin oligodendrocyte glycoprotein (MOG). The administration of AAV with CD40-Ig protected mice from EAE and altered disease progression [69,70].

In vivo studies in a mouse model of EAE induced by MOG demonstrated the prophylactic and therapeutic efficacy of the continuous delivery of interferon alpha 1 (IFN-α1), alone or in combination with apolipoprotein A-1 via the AAV system [68,71]. Moreover, the long-term administration of low doses of IFN-α produced immunomodulatory effects, including the expansion of pro-inflammatory monocytes and regulatory T cells [68,71]. Another approach is to use the herpes simplex virus to knock out genes encoding cytokines. In studies on mice, this genetic vector inhibited the development of MS and prevented disease progression [72]. The use of LVs carrying a gene encoding proteins with anti-inflammatory or neuroprotective effects also has therapeutic potential for MS [73]. However, these studies are still in the preclinical phase, and their efficacy and safety require further investigation. Nevertheless, the use of immunotherapy using viral vectors offers many possibilities for the treatment of MS.

### 4.4. Glioblastoma Multiforme

GBM is the most common malignant brain tumor in adults. It has a high mortality rate, with a 5-year survival rate of 7.2% [74]. GBM is a complex disease with many variables, including its location in the brain, cellular origin, age, gender, and tumor subtype [75]. Current treatments for GBM include the surgical resection of the tumor and chemoradiotherapy with temozolomide [76,77,78,79]. Despite aggressive therapy and maximal tumor removal, the tumor is susceptible to local recurrence. This is due to its distinct features, such as its infiltrative nature, high degree of heterogeneity, immunosuppressive tumor microenvironment, and the presence of the BBB [80,81]. The infiltrative nature of the tumor prevents complete resection, and the high heterogeneity makes targeted therapy much more difficult [74].

These limitations, due to the characteristics of GBM, have led researchers to explore new treatments. One of the viruses being clinically tested for the treatment of GBM is vocimagene amiretrorepvec (Toca 511), which is based on a nonlytic strategy [74].

Toca 511 is a replication-competent retroviral vector derived from a modified mouse leukemia virus. The vector carries a transgene for modified yeast cytosine deaminase, which significantly improves genome stability and enhances the enzyme’s activity compared to its original form [82]. The therapy involves the initial resection and administration of a replicating retrovirus into the tumor locus to infect tumor cells for a specified period of time (several weeks). The injected virus converts the prodrug, Toca FC (contains 5-fluorocytosine), into the cytotoxic 5-fluorouracil, which destabilizes thymidylate synthase and disrupts cell replication [74,78]. When viral infection occurs, the enzyme that transforms the prodrug can convert valacyclovir into a nucleotide analog so that cancer cell proliferation is interrupted [74,78,80]. Additionally, Toca 511 can infect only actively dividing cells, so the virus remains selective and infects cancer cells [82].

The other promising alternative is suicide gene therapy against glioma, which uses a system of enzymes and drugs to inhibit tumor growth [75]. Genetically modified viral vectors introduced into the body allow the prodrug to convert into toxic metabolites in the targeted tumor cells, causing their death. The most extensively studied suicide gene therapy is one that uses herpes simplex virus type 1 thymidine kinase [75]. This enzyme induces cell cytotoxicity by catalyzing the phosphorylation of nucleoside analogs, such as gancyclovir and acyclovir, leading to the formation of toxic gancyclovir triphosphate and acyclovir triphosphate [75].

Some clinical trials using a viral vector as a drug delivery system are shown in Table 3, which was generated from the form https://clinicaltrials.gov (accessed on 5 April 2023).

### 4.5. Canavan Disease

CD is a lethal monogenic, autosomal recessive white matter disease caused by loss of-function mutations in the *ASPA* gene encoding aspartoacylase. The abnormal function of the enzyme leads to increased amounts of N-acetyl-l-aspartate (NAA) molecules in the CNS. The result is impaired normal myelination, vacuolization of the white matter, and midline swelling, leading to hydrocephaly [83,84,85,86,87]. Currently, no effective treatments are available for this condition. Available drug therapies are limited to improving patients’ quality of life by suppressing epileptic seizures and orally administering lithium acetate, lithium citrate, and other agents such as glycerol triacetate [86]. However, virus-based gene transfer can halt or reverse the course of the disease [83,84].

Based on preclinical studies of Canavan-like diseases, it has been found that these pathologies can be treated with *ASPA* gene replacement therapy or by lowering the expression of the enzyme that synthesizes NAA [83]. The insertion of a healthy copy of the *ASPA* gene into the patients’ cells is performed using a modified virus. The first preclinical studies of gene therapy to combat Canavan’s disease were conducted in healthy rodents and primates via intracerebral injections [83]. A delivery system surrounded by polycations and lipids was used in conjunction with AAV. The efficacy of the concept was tested on two children. The effect of gene therapy was well tolerated and some biochemical and radiological parameters improved, but no clinically significant improvement in the course of the disease was observed [85,88].

A few years later, a study complementing earlier theories was conducted on a larger group of patients with a modified, improved AAV2-ASPA enzyme delivery system [84]. Further studies toward this conclusion have shown that AAV2-ASPA gene therapy slows the progression of brain atrophy, reduces the number of epileptic seizures, and stabilizes the overall clinical status. Studies have also suggested that the age of the patient affects the success of the therapy. The younger the patient, the more significant the improvement in health [89].

In 2023, ten studies on CD were noted, three of which used gene therapy linked to the viral carrier AAV. Relevant information has been collected from the https://clinicaltrials.gov database (accessed on 3 March 2023) and is presented in Table 4.

CD gene therapies using recombinant adeno-associated viruses (rAAVs) are a promising method of gene delivery due to their broad tissue tropism, low immunogenicity, high efficiency, and sustained gene transduction [90]. In addition to AAV, other types of viruses are being used, one of which is a viral vector based on the LV. Recent studies have combined gene therapy with cell therapy. They used modified induced pluripotent stem cell (iPSC) cells derived from a CD patient with the expression of the *WT ASPA* gene (ASPA-CD iPSC) via lentiviral transduction and demonstrated their therapeutic efficacy and preliminary safety in a mouse model [87,91]. Subsequent studies have suggested that LVs are likely to be more stable due to integration events, resulting in sustained ASPA activity in the patient’s brain, unlike transgenes provided by AAVs [92].

## 5. Virus-like Particles in the Treatment of Central Nervous System Diseases

### 5.1. Properties of Virus-like Particles

Advances in nanotechnology and pharmaceuticals are contributing to the emergence of new developments in drug technology. Therefore, in addition to the current drug delivery systems based on inorganic and organic materials, as well as virus-based systems, there is an increasing focus on the targeted delivery of molecules (e.g., doxorubicin, paclitaxel) for CNS disease therapy using virus-like particles (VLPs) [93,94,95].

VLPs are classified as self-assembling protein-based nanoparticles with a size range of 20–200 nm, rendering them capable of crossing the BBB, which is highly significant in the treatment of CNS diseases [96,97]. A characteristic feature of VLPs is the fact that, in addition to the capsid proteins, they possess only the genetic material required for the replication process without causing harmful infection [98]. Occasionally, such particles are completely devoid of genetic material [98].

The morphology, structure, packaging, and properties of VLPs vary and depend primarily on the initial form of the virus they mimic, as well as their biomedical application (Figure 5) [83].

In addition, some VLPs derived from viruses exhibit natural tropism to central nervous system cells [98,99]. One example is the John Cunningham polyomavirus, whose prototype form is responsible for oligodendrocyte infections and contributes to CNS diseases, which makes it a strong candidate as a drug carrier in brain-related diseases [99]. An additional advantage of using this type of drug delivery system is its ability to functionalize the surface of the VLP for targeted drug delivery. For this purpose, chemical or genetic engineering methods are mainly used to conjugate with antibodies or other biomolecules, such as ferritin, thus improving bioavailability [96,100]. Because VLPs are biodegradable and exhibit relatively low toxicity, they are officially approved by the Food and Drug Administration (FDA) for use as drug carriers [101]. VLPs have been used in studies of drug delivery (e.g., paclitaxel, doxorubicin), proteins (e.g., epitopes), nucleic acids (e.g., siRNA, mRNA), enzymes (e.g., AADC), and as vaccines (e.g., against malaria). They are also used in diagnostics [102,103].

### 5.2. Morphological Forms of Virus-like Particles

VLPs, as nanometric protein structures that lack infectious genetic material, are formed by the spontaneous self-assembly reaction of individual proteins [19,101]. This results in the formulation of vectors that are structurally and visually similar to the viruses from which they originate. In this process, VLP particles can take four geometric forms of the capsid at the construct assembly stage: icosahedral, spherical, disc-shaped, and rod-shaped [19,101]. Moreover, complex structures that combine various shapes can also be formed [96,104]. In addition to morphological forms, they can also be differentiated in terms of the number of capsid-building layers, where monolayers, bilayers, and trilayers with a minimum of one capsid protein are distinguished [19,105]. Also important for the drug delivery and production of VLPs is the absence or presence of lipid envelopes, obtained mainly from the lipid membrane of host cells [19,105,106].

Nonenveloped VLPs do not have the aforementioned envelope in their structure [19,105]. The main advantage is that the method of obtaining and purifying VLPs is significantly simpler than that of VLPs with envelopes. In addition, the size is smaller, resulting in the ability to overcome natural barriers in the body, such as the BBB [19,105,106]. They consist of a single capsid protein assembled in eukaryotic or prokaryotic expression platforms [19]. Nonenveloped VLPs can also be formed from a multi-protein capsid using more advanced methods involving eukaryotic cells, such as plants or mammalian cells, as the expression platform [19,105,107]. The use of multiprotein nonenveloped VLPs is one of the most widely used VLP systems in drug delivery. In this class of VLPs, we can include bluetongue virus, the L1 and L2 proteins of the human papilloma virus, and poliovirus [19].

Enveloped VLPs (eVLPs) have a lipid envelope derived from the host they infect. They can also be multilayered and contain several different capsid proteins [19,105]. However, they are much less commonly used in drug delivery systems because of severe storage problems [107]. Among other reasons, this is due to the fact that they have a lipid envelope, which is extremely sensitive to the thermal conditions and shear forces generated during the purification stage [107]. For this reason, VLPs without the envelope are gaining more attention in therapy, where their use has been reported in research for the treatment of colorectal cancer and CNS-related diseases, among others [19,101,108,109].

### 5.3. Expression Platforms Used in the Production of Virus-like Particles

The production process and the selection of a suitable expression system for the desired VLP are crucial [19,107]. First and foremost, the purpose of the VLP application must be considered, as well as the requirements necessary for protein folding and the possibility of post-translational modifications for specific proteins [104]. Generally, five expression systems are distinguished: bacterial, yeast, plant, insect, and mammalian cells (Figure 6) [19,107,110].

For the production of VLPs as carriers of medicinal substances for the treatment of CNS diseases, those based on bacterial, plant, and mammalian systems are primarily used [101].

Bacterial expression systems are among the most widely used platforms for generating VLPs [19,106]. This is due to their relatively low production cost and simplicity of manufacturing. They are most suitable for producing nonenveloped VLPs, which, as described earlier, are most commonly used in drug delivery systems [19]. The disadvantages of this method, however, are problems with the production of eVLPs due to the lack of a post-translational protein modification system in bacterial cells [19,106,107]. Leading the way in this system are the bacteria *Escherichia coli*, *Lactobacillus casei*, and *Pseudomonas fluorescens*, whose doubling time is relatively short, making the process of VLP production potentially shorter [19].

Eukaryotic cells, i.e., plant and mammalian cells, are also used to produce a VLP that can be used as a nanocarrier for the medicinal substance. Using plants as nanoreactors for VLPs is a more efficient system than using bacteria because it is possible to obtain a large amount of soluble bilayer at reduced production costs [19,107,111]. Several plants have found application in CNS diseases, including *Nicotiana tabacum* and *Nicotiana benthamiana* [19,111]. Expression systems using mammalian cells are used due to the possibility of generating all forms of VLPs, both nonenveloped and eVLPs, owing to the highly developed post-translational modification system occurring in these cells [19,107]. In addition, by using such a system to produce VLPs, it is possible to create chimeric VLPs; furthermore, there is the possibility of functionalizing the surface of VLPs to increase the duration of the particle’s residence in the body or the targeted delivery of medicinal substances. Among the most commonly used cell lysates are human embryonic kidney 293 (HEK293T), Chinese hamster ovary (CHO), and baby hamster kidney-21 (BHK-21) [19,106,107].

### 5.4. Application of VLPs as Drug Delivery Systems in the Treatment of CNS Diseases

VLPs, due to their distinctive properties, i.e., small size (20–200 nm), ability to carry medicinal substances to a specific site, ease of surface functionalization, and natural tropism to cells while being biodegradable and low in toxicity, have found application in CNS disease therapy research [101]. Finbloom et al. [101] investigated the potential of delivering doxorubicin (DOX) using VLPs administered via CEDs in the treatment of GBM as an alternative to administering the native form of this chemotherapeutic agent [101]. In the study, they tested three morphologically different forms of VLP, i.e., MS2 bacteriophage spheres, tobacco mosaic virus (TMV) discs, and filamentous nanophage rods that each contained DOX [101]. They used the U-87 MG cell line as their test model, and also carried out experiments on mice bearing intracranial tumors. Using a CED infusion cannula, the corresponding morphological forms of VLP-DOX were injected intracranially at a concentration of 20 µg/kg [101]. The results of the study showed the increased survival of mice after the first administration of DOX conjugated to TMV discs and MS2 spheres, suggesting a more effective dosing of the chemotherapeutic agent at a lower dose compared to the intravenous administration of nonVLP-conjugated doxorubicin [101]. This study highlights the undoubted potential of VLPs as carriers of medicinal substances, thus limiting the administration of higher doses of the drug.

Additional examples of the application of VLPs as drug delivery systems for the treatment of CNS diseases are summarized in Table 5.

## 6. Industrial Production, Regulatory Requirement and Limitations of Using Virus-Based Biological Systems as Carriers for Therapy

The process of producing viral carriers on a large scale usually involves introducing a vector structure containing the desired gene and the necessary, e.g., lentiviral elements, such as regulatory and structural sequences, into packaging cells such as HEK293T [73]. After a few days, the packaging cells begin to produce particles of the desired vector, which can later be harvested from the culture medium [73]. This is followed by a purification process, which, depending on the intended use, may involve the use of various methods, such as gradient purification or chromatography. After this process, the eluted fractions undergo a series of filtrations to sterilize and remove any cellular residues. Such carriers can be stored for up to 9 years if cryopreserved at −80 °C [73].

A major problem in the production of new therapies based on viral vectors, especially lentiviruses and retroviruses, is the possibility of spontaneous recombination during the production process in packaging cells [73,122]. This may result in the formation of progenitor viruses that are able to replicate in host cells, which is directly related to the possibility of the development of viral disease in the treated patient. Therefore, to prevent the commercialization of such formulations, the FDA requires testing for the presence of replication-capable retroviruses and lentiviruses (RCR/RCL) to be performed on both cell lines and on the formulation after the purification step [73,122]. It is recommended that patients using preparations based on viral vectors be monitored for the presence of RCR/RCL using molecular biology methods, i.e., polymerase chain reaction [73]. In addition, when treating cancer with viral vectors, it is advisable to monitor patients against secondary malignancies. During treatment, the patient should be observed for the exacerbation of already existing diseases, especially autoimmune, neurological, and rheumatoid diseases. To properly treat formulations containing viral elements, the patient needs to be acquainted with the mechanism of action of a given product and should be informed about the possible consequences of integrating the viral genome into DNA [73,122].

It is worth mentioning the limitations of viral vectors and VLPs. Viral vectors and nonviral particles, by virtue of their biology, require host cellular machinery for their self-organization. This phenomenon makes the production of these vectors difficult due to the long, tedious process of cell culture and the risk of contamination and loss of raw material [73,122]. Because the described transporters are protein creations containing genetic material, their problems are their short-term stability and the difficulties experienced when storage them [73]. It is also noteworthy that not all serotypes of viruses are capable of crossing the BBB; only some viruses are capable of this, and it is necessary to search for and synthesize an increasing number of new serotypes to enhance their ability to cross biological membranes. The problem mentioned regarding the use of viruses mainly concerns the treatment of MS [72]. This is because the pathogenesis of this disease is highly associated with HSV-1 virus infection, which causes demyelination and the occurrence of disease symptoms [72]. In a study on mice, researchers used HSV-1 with a deletion of the *ICP34.5* gene, which is responsible for neurovirulence. The virus used with the deletion of *ICP34.5* showed a reduced ability to replicate and reactivate in the nervous system. However, there are studies showing that viruses with *ICP34.5* deletion can cause neuronal destruction and produce a strong immune response in the brains of various strains of mice, which is associated with significant weight loss in symptomatic animals [72]. These results suggest that the use of these virus mutants may be ineffective and may have negative effects on the treatment of MS patients [72]. Therefore, the use of other types of viruses, such as AAV, in the treatment of MS is necessary.

Further research is also needed on tropism, i.e., the ability to precisely deliver molecules to appropriate tissues and cells, and the development of VLPs specific to target cells, as well as controlled cargo release. This research can greatly enhance the effectiveness of the method and remains one of the main areas for improvement [122].

However, in addition to these limitations, new patents are being registered for modifications of viruses and VLPs as carriers of medicinal substances, mainly based on their functionalization or the process of obtaining mutants and new serotypes. Therefore, several patents related to the production of viruses and VLPs as carriers are presented in Table 6, generated from the European Patent Office, https://epo.org (accessed on 18 June 2023).

## 7. Conclusions

This review presents numerous studies based on the use of viral vectors and virus-like particles in CNS disease therapies. In the case of viral vectors, it is possible to modify capsid proteins, create chimeras, or manipulate the capsid surface to achieve the desired tissue tropism or migration across biological barriers, such as the BBB. With VLPs, the possibilities are even greater, as it is possible to transport chemotherapeutics, manipulate the shape and size of the vector, and modify its structure. Also, the advantages of viral vectors include their ability to express proteins or genes over the long term and their safety due to the low incidence of side effects and their ability to manipulate the immune profile. Viral vectors and VLPs are successfully used in gene therapies because they effectively protect the transgene and arming them with appropriate promoters facilitates their integration and expression in target cells. Due to the simplicity of viral genome modification, we can easily introduce a large group of transgenes into the aforementioned vectors. The properties of penetration through biological barriers or particle sizes allow them to reach their targets in high concentrations. However, some limitations of viral vectors and virus-like particles should not be forgotten.

The data presented in this work indicate that viral vectors and VLPs are a promising strategy for delivering therapeutic agents in CNS diseases. Nevertheless, to strive for the most effective delivery of medicinal substances, further research on these vectors is needed to improve their tropism, stability, and ability to cross biological barriers or refine viral delivery methods.

## Figures and Tables

**Figure 1 pharmaceutics-15-01931-f001:**
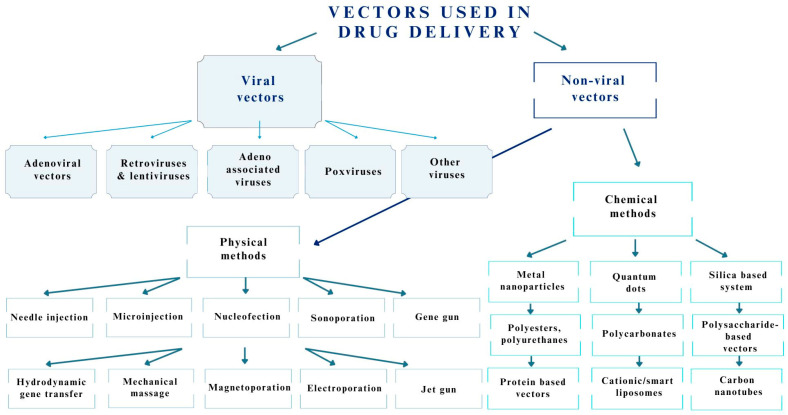
Classification of vectors used in drug delivery systems in the treatment of CNS diseases.

**Figure 2 pharmaceutics-15-01931-f002:**
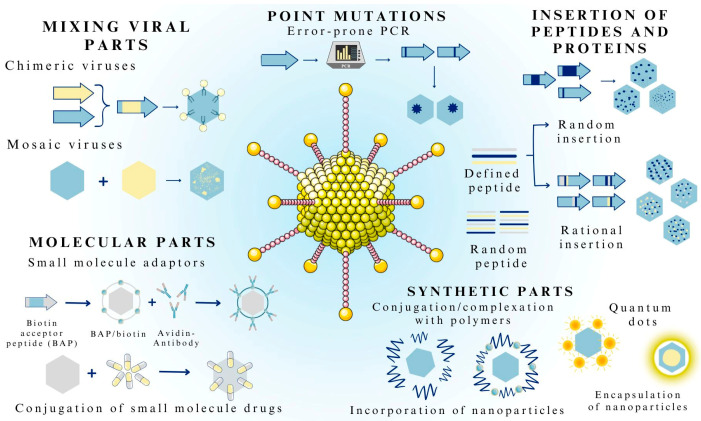
Methods of formulating viruses as carriers of medicinal substances. The figure was partly generated using Servier Medical Art, provided by Servier and licensed under a Creative Commons Attribution 3.0 unported license.

**Figure 3 pharmaceutics-15-01931-f003:**
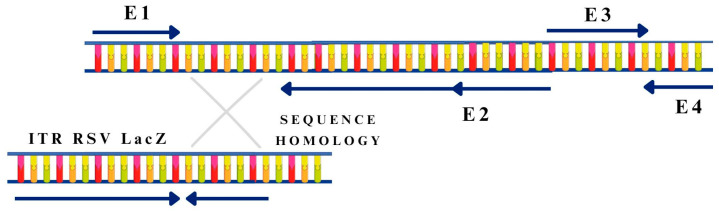
Scheme for obtaining recombinant viruses as gene carriers. The figure was partly generated using Servier Medical Art, provided by Servier and licensed under a Creative Commons Attribution 3.0 unported license.

**Figure 4 pharmaceutics-15-01931-f004:**
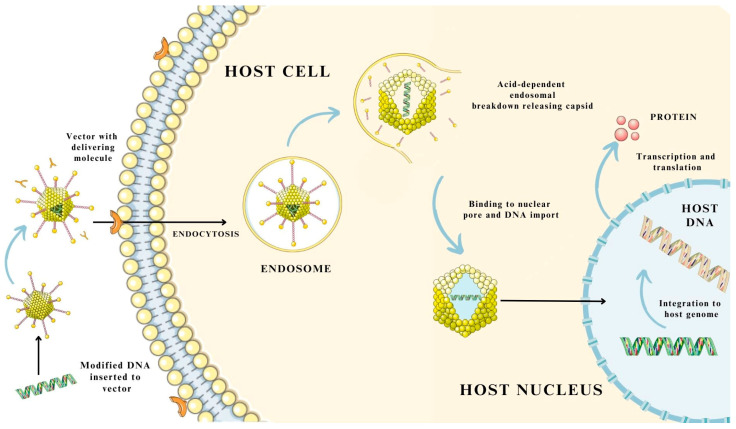
Example of a drug delivery mechanism via viruses and virus-like particles. The figure was partly generated using Servier Medical Art, provided by Servier and licensed under a Creative Commons Attribution 3.0 unported license.

**Figure 5 pharmaceutics-15-01931-f005:**
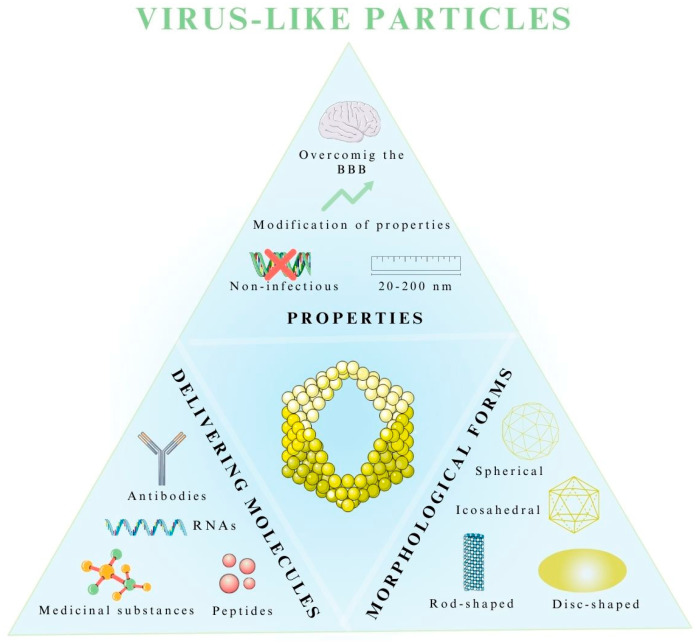
Properties, morphological forms, and delivering molecules of virus-like particles as drug delivery systems. The figure was partly generated using Servier Medical Art, provided by Servier and licensed under a Creative Commons Attribution 3.0 unported license.

**Figure 6 pharmaceutics-15-01931-f006:**
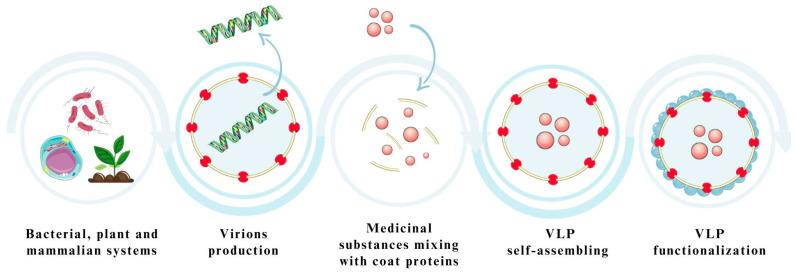
Methods for producing VLPs using different expression platforms. The figure was partly generated using Servier Medical Art, provided by Servier and licensed under a Creative Commons Attribution 3.0 unported license.

**Table 1 pharmaceutics-15-01931-t001:** Application of viruses as drug delivery systems for the treatment of selected CNS diseases.

Viral Drug Delivery System	Type of Virus	Tissue Targeting	Molecule	In vitro/In vivo	Disease	Ref.
pLentiM1.2-hNRGN	LV	Hippocampus cells	Ng	In vivo—C57BL/6 mice; 5XFAD mice	AD	[38]
Recombinant adenovirus	AV	Malignant glioma cells	shRNA RHBDD1	In vitro—U87MG; U251	GBM	[39]
Ad5scFvDEC205FF	AV	Dendritic cells	scFv	In vitro—GL261; GL261CMV-IE In vivo—C57BL/6 mice	GBM	[40]
Ad-CALR/MAGE-A3	AV	Glioma cells	CALR; MAGE-A3	In vitro—U87MG	GBM	[41]
AAV.SIRT3-myc	AAV	Stratum	SIRT3	In vivo—Sprague Dawley rats	PD	[42]
AAVrh.10hAPOE2-HA	AAV	Hippocampus cells	APOE2	In vivo—adult *Chlorocebus aethiops sabaeus* NHPs	AD	[43]
Modified AAV	AAV	Cells of the substantia nigra	TRPV4 shRNAi	In vivo—C57BL/6J mice	PD	[44]

pLentiM1.2-hNRGN—lentiviral vector expressing neurogranin under the control of the mouse cytomegalovirus immediate-early promote; LV—lentivirus; Ng—neurogranin; AD—Alzheimer’s disease; AV—adenovirus; shRNA RHBDD1—short hairpin RNA targeting rhomboid domain containing 1; GBM—glioblastoma multiforme; Ad5scFvDEC205FF—human adenovirus serotype 5 single-chain variable fragment against dendritic and epithelial cell receptor with a molecular weight of 205 kDa incorporated into chimeric fiber fibritin; scFv—single-chain variable fragment; Ad-CALR/MAGE-A3—adenovirus delivering calreticulin and melanoma antigen-A3 genes; CALR—calreticulin; MAGE-A3—melanoma antigen gene-A3; AAV.SIRT3-myc—adeno-associated virus expressing myc-tagged sirtuin 3; AAV—adeno-associated virus; SIRT3—sirtuin 3; PD—Parkinson’s disease; AAVrh.10hAPOE2-HA—adeno-associated virus rhesus isolate 10 serotype coding for an HA-tagged human apolipoprotein E allele 2 sequence; APOE2—apolipoprotein E allele 2; NHPs—nonhuman primates; TRPV4 shRNAi—transient receptor potential vanilloid 4 short hairpin RNA interference.

**Table 2 pharmaceutics-15-01931-t002:** Selected clinical trials of virus-based gene delivery therapy for Alzheimer’s disease and Parkinson’s disease found at https://clinicaltrials.gov (accessed on 3 April 2023).

Research Title	Drug/Molecule	Status	Application Route	NCT Number	Phase/Disease	Participants
Randomized, Controlled Study Evaluating CERE-110 in Subjects with Mild to Moderate Alzheimer’s Disease	CERE-110, (AAV2-NGF)	Non recruiting	Injection into brain	NCT00876863	II/AD	49
Gene Therapy for APOE4 Homozygote of Alzheimer’s Disease	LX1001(AAVrh.10hAPOE2)	Recruiting	n/d	NCT03634007	I and II/AD	15
AAV2-GDNF for Advanced Parkinson’s Disease	AAV2-GDNF	Non recruiting	Injection into single arm	NCT01621581	I/PD	25
A Study of AAV-hAADC-2 in Subjects with Parkinson’s Disease	AAV-hAADC-2	Non recruiting	Injection into the striatum	NCT00229736	I/PD	10
Study of AAV-GAD Gene Transfer into the Subthalamic Nucleus for Parkinson’s Disease	AAV-GAD	Terminated (due to financial reasons)	Infusion into the subthalamic nucleus region of the brain	NCT00643890	II/PD	44

AAV2-NGF—adeno-associated virus serotype 2 delivering nerve growth factor gene; AD—Alzheimer’s disease; AAVrh.10hAPOE2—adeno-associated virus rhesus isolate 10 serotype coding apolipoprotein E allele 2 sequence; n/d—no data; AAV2-GDNF—adeno-associated virus serotype 2 containing the human glial cell line-derived neurotrophic factor; PD—Parkinson’s disease; AAV-hAADC-2—adeno-associated virus serotype 2 encoding human aromatic l-amino acid decarboxylase; AAV-GAD—adeno-associated virus delivering glutamic acid decarboxylase gene.

**Table 3 pharmaceutics-15-01931-t003:** Selected clinical trials of gene therapy for glioblastoma multiforme found at https://clinicaltrials.gov (accessed on 5 April 2023).

Research Title	Drug/Molecule	Status	Application Route	NCT Number	Phase	Participants
Study of a Retroviral Replicating Vector Combined with a Prodrug to Treat Patients Undergoing Surgery for a Recurrent Malignant Brain Tumor	Toca 511; Toca FC	Non recruiting	Injections into resection cavity wall; orally	NCT01470794	I	56
The Toca 5 Trial: Toca 511 & Toca FC Versus Standard of Care in Patients with Recurrent High Grade Glioma (Toca5)	Toca 511; Toca FC	Non recruiting	Injections into resection cavity wall; orally	NCT02414165	II/III	403
Viral Therapy in Treating Patients with Recurrent Glioblastoma Multiforme	MV-CEA	Completed	Injection into resection cavity or around tumor bed	NCT00390299	I	23
DNX-2440 Oncolytic Adenovirus for Recurrent Glioblastoma	DNX-2440	Completed	Injection stereotactically	NCT03714334	I	16
Genetically Engineered HSV-1 Phase 1 Study for the Treatment of Recurrent Malignant Glioma (M032-HSV-1)	M032	Active, not recruiting	Infusion through catheters into regions of tumor	NCT02062827	I	24
PVSRIPO for Recurrent Glioblastoma (GBM) (PVSRIPO)	PVSRIPO	Completed	Infusion into the tumor	NCT01491893	I	60

Toca 511—vocimagene amiretrorepvec; Toca FC—drug containing Toca 511 vector in combination with 5-fluorocytosine; MV-CEA—carcinoembryonic antigen-expressing measles virus; M032—second-generation oncolytic herpes simplex virus enhancing secretion of interleukin-12; PVSRIPO—recombinant nonpathogenic polio–rhinovirus chimera.

**Table 4 pharmaceutics-15-01931-t004:** Selected clinical trials of gene therapy for Canavan disease found at https://clinicaltrials.gov (accessed on 3 March 2023).

Research Title	Drug/Molecule	Status	Application Route	NCT Number	Phase	Participants
A Study of AAV9 Gene Therapy in Participants with Canavan Disease (CANaspire)	AAV9 BBP-812	Recruiting	Intravenous infusion	NCT04998396	I and II	18
rAAV-Olig001-ASPA Gene Therapy for Treatment of Children with Typical Canavan Disease (CAN-GT)	rAAV-Olig001-ASPA; levetiracetam; prednisone	Active non recruiting	Intraventricular; administered orally or by gavage	NCT04833907	I and II	24
Canavan-Single Patient IND	rAAV9-CB6-AspA	Available	Single intravenous and intraventricular	NCT05317780	n/d	1

AAV9 BBP-812—adeno-associated virus serotype 9 delivering the aspartoacylase transgene; rAAV-Oligo001-ASPA—recombinant adeno-associated virus selectively targets oligodendrocytes delivering the aspartoacylase transgene; rAAv9-CB6-AspA—recombinant adeno-associated virus serotype 9 carrying the human aspartoacylase gene with a modified cytomegalovirus-enhancer chicken β-actin (CB6) promoter; n/d—no data.

**Table 5 pharmaceutics-15-01931-t005:** Application of VLPs as a drug delivery system for the treatment of selected CNS diseases.

Virus-Like Particle	Virus	Molecule	Expression Platform	In vitro/In vivo	Disease	Ref.
HBc VLPs	*Hepatitis B virus*	tau_294–305_ protein	*Escherichia coli* (BL21)	In vivo—Tau.P301S mice	AD	[112]
Qβ VLPs	*Qβ bacteriophage*	pT181	*Escherichia coli*	In vivo—bitransgenic rTg4510 mice	AD	[113]
HPV16 L1a and L1b VLPs	*Human papilloma virus*	Aβ_11–28_ epitope	Plants	In vivo—C57BL/6J mice	AD	[114]
CMV VLPs	*Cucumber mosaic virus*	N-terminus of Aβ_1–42_ epitope	n/d	In vivo—C57BL/6J mice; BALB/c mice	AD	[115]
HIV-2 and SIV VLPs	*Human immunodeficiency virus type 2* and *simian immunodeficiency virus*	Vpx	293T cells	In vitro—H4; LN-229; U87 MG In vivo—female athymic nude mice	GBM	[116]
JCPyV VLPs	*JC polyomavirus*	GFP and thymidine kinase suicide gene	*Escherichia coli* (JM109)	In vitro—U87 MG In vivo—nu/nu mice	GBM	[117]
Qβ VLPs with surface modification by CCP and ApoE	*Qβ bacteriophage*	RNAi_c-met_	*Escherichia coli*	In vitro—U87 MG	GBM	[118]
BTV VLPs	*Bluetongue virus*	HSV1-TK	*Nicotiana benthamiana*	In vitro—U87 MG	GBM	[119]
TGN/RGD-HBc VLPs	*Hepatitis B virus*	Co-delivery: Paclitaxel and YAP siRNA	*Escherichia coli* (BL21)	In vitro—U87 MG In vivo—BALB/c mice	GBM	[120]
Qβ VLPs with human a-syn	*Qβ bacteriophage*	Synthetic peptides: ○ GKNEEGAPQ ○ MDVFMKGLGGC ○ CGGEGYQDYEPEA	*Escherichia coli* (JM109)	In vivo—C57BL/6 mice	PD	[121]

HBc—hepatitis B virus core protein; AD—Alzheimer’s disease; VLPs—virus-like particles; pT181—tau peptide phosphorylated at threonine 181; HPV16—human papillomavirus type 16; CMV—cucumber mosaic virus; n/d—no data; HIV-2—human immunodeficiency virus type 2; SIV—simian immunodeficiency virus; Vpx—viral protein X; GBM—glioblastoma multiforme; JCPyV—John Cunningham polyomavirus; GFP—green fluorescent protein; CCP—cell-penetrating peptide; ApoE—apolipoprotein E peptide; RNAi_c-met_—RNA interference tyrosine-protein kinase; BTV—bluetongue virus; HSV1-TK—herpes simplex virus type 1 thymidine kinase; TGN—12-amino acid peptide (TGNYKALHPHNG); RGD—arginine–glycine–aspartic acid; YAP—yes-associated protein; siRNA—small interfering ribonucleic acid; a-syn—alpha synuclein; PD—Parkinson’s disease.

**Table 6 pharmaceutics-15-01931-t006:** Examples of recent patents and formulations using virus-based systems as drug carriers for disease therapy are found at https://epo.org (accessed on 18 June 2023).

Patent Title	Patent Number	Viral Vector	Molecule/Drug	Disease	Brief Description
Compositions and methods for the treatment of neurological disorders related to glucosylceramidase beta deficiency	WO2023091949A2	AAV particle	Gene encoding GCase	PD	Increased delivery of the gene encoding GCase to enhance alleviation of loss of function and intracellular lipid transport. Improved lysosomal glycolipid metabolism results in reduction, arrest or reversal of PD symptoms
Compositions and methods for the treatment of tau-related disorders	WO2023092004A1	AAV particle	Antibody molecule binding to tau	AD	Delivery of antibodies targeting tau proteins via AAV viral particles, which have a modified genome. Such modification enables expression of anti-tau antibodies by the virus, contributing to the treatment of diseases associated with tau protein pathology
AADC/GDNF polynucleotide, and use thereof in treating Parkinson’s disease	WO2023093905A1	AAV virus	AADC/GDNF polynucleotide	PD	Utilization and optimization of AAV vector to deliver genes encoding AADC and GDNF in the form of polynucleotides for the treatment of Parkinson’s disease patients
A method for providing a VLP derived from John Cunningham virus	CA3121689A1	JCV VLP	Proteins	Brain-related diseases	The invention relates to methods for the delivery of JCV VLPs containing a protein cargo, in particular, the development of a step for the disassembly of VLPs, their re-aggregation and the formation of VLPs. This is aimed at developing an optimal method of drug delivery to areas where the BBB is an obstacle

AAV—adeno-associated virus; GCase—β-Glucocerebrosidase; PD—Parkinson’s disease; AD—Alzheimer’s disease; AADC—aromatic l-amino acid decarboxylase; GDNF—glial cell-derived neurotrophic factor; VLP—virus-like particle; JCV—John Cunningham virus; BBB—blood–brain barrier.

## Data Availability

Available on request and with regulations.

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
