# Peer review of "Virus-Based Biological Systems as Next-Generation Carriers for the Therapy of Central Nervous System Diseases"

_pharmaceutics, 2023, doi:10.3390/pharmaceutics15071931_

Round 1

Reviewer 1 Report

The Review "Virus-Based Biological Systems as Next Generation Carriers for the Therapy of Central Nervous System Diseases" present recent literature in the field of utilization of viral vectors and VLPs 21 in the treatment of selected CNS diseases. The topic is related to many current research work and novel. The review covered, Limitations of the Current drug delivery system to CNS and the application of viruses as a drug delivery system for the treatment of selected CNS diseases including various clinical trial data. The comments to improve the manuscript are as follows: 

1. L: 37, Citation should be like [6-9], not [6,7,8,9]

2. The limitation discuss in the conclusion, please provide a separate paragraph before the conclusion with some literature support. Expecting industrial production too. 

3. Please provide regulatory requirements or concerns. 

4. Patents and marketed formulations if any list a table and discuss in brief. 

Author Response

Review 1:

Comments:

(x) I am not qualified to assess the quality of English in this paper
( ) English very difficult to understand/incomprehensible
( ) Extensive editing of English language required
( ) Moderate editing of English language required
( ) Minor editing of English language required
( ) English language fine. No issues detected

Is the work a significant contribution to the field?

4/5

Is the work well organized and comprehensively described?

4/5

Is the work scientifically sound and not misleading?

4/5

Are there appropriate and adequate references to related and previous work?

4/5

Is the English used correct and readable?

4/5

Answer:

Our work has undergone major modifications. We tried to take into account all the comments of the Reviewer increasing the quality of our work.

Comment:

“1. L: 37, Citation should be like [6-9], not [6,7,8,9]”.

Answer:

Thank you to the Reviewer for the right comment. The citation notation throughout the manuscript has been corrected.

Comment:

2. The limitation discuss in the conclusion, please provide a separate paragraph before the conclusion with some literature support. Expecting industrial production too”.

Answer:

We thank the Reviewer for the suggestion, which will improve the quality of the work. Therefore, we have introduced a paragraph demonstrating the limitations of the use of virus-based systems and also in this paragraph we have included information on regulations, industrial production along with a table with the latest patents in this field.

Comment:

“3. Please provide regulatory requirements or concerns”. 

Answer:

Regulatory requirements were added as suggested by the Reviewer and are presented in a separate paragraph.

Comment:

“4. Patents and marketed formulations if any list a table and discuss in brief”. 

Answer:

We would like to thank the Reviewer for his valuable suggestion, according to which we introduced a table with the latest patents in the field of the subject discussed.

Reviewer 2 Report

I suggested that the limitations of virus-based biological systems as carriers for therapy central nervous system should be added in the MS.

Author Response

Review 2:

Comments:

( ) I am not qualified to assess the quality of English in this paper
( ) English very difficult to understand/incomprehensible
( ) Extensive editing of English language required
( ) Moderate editing of English language required
( ) Minor editing of English language required
(x) English language fine. No issues detected

Is the work a significant contribution to the field?

4/5

Is the work well organized and comprehensively described?

4/5

Is the work scientifically sound and not misleading?

4/5

Are there appropriate and adequate references to related and previous work?

4/5

Is the English used correct and readable?

4/5

Answer:

Our work has undergone major modifications. We tried to take into account all the comments of the Reviewer increasing the quality of our work.

Comment:

“I suggested that the limitations of virus-based biological systems as carriers for therapy central nervous system should be added in the MS”.

Answer:

We thank the Reviewer for the accurate comment. Restrictions on the application of the discussed carriers in MS therapy are presented in a newly added paragraph (6. Industrial Production, Regulatory Requirement and Limitations of Using Virus-Based Biological Systems as Carriers for the Therapy).

Reviewer 3 Report

While I appreciate the effort and thought put into the this study, I believe it falls in a unique position that may not cater to the needs of either beginner or advanced researchers in the field. On one hand, the manuscript does not provide a sufficiently general and comprehensive overview suitable for beginners in the field. It lacks a broader context and fails to delve into fundamental concepts that would enable newcomers to grasp the subject matter more easily. As a result, novice researchers might struggle to fully understand the content without prior foundational knowledge. On the other hand, the manuscript does not offer the level of detail, depth, and novelty that would engage and benefit advanced researchers in the field. While it covers certain aspects reasonably well, it does not explore more intricate or cutting-edge areas that could contribute to the existing body of knowledge.

While the manuscript presents valuable insights, there are significant issues related to consistency, readability, and suitability of the English writing that need to be addressed. 

In relation to Figure 4, it is advised that the authors make adjustments to enhance reader comprehension. Initially, the left-side cell membrane implies a misconception regarding virus particle exocytosis and cell release, which is not accurate. By enhancing the figure, the process of virus particle endocytosis can be presented in a more lucid manner.

Regarding the language and grammar, it is recommended to the authors to thoroughly proofread the article to correct grammatical errors, improve sentence structure, and ensure proper word usage. Few examples of language and grammatical errors are provided below: 

- Line 89: the sentence 'Another promising concept are also ...' contains potential grammatical error that needs to be corrected. 

- Line 137: at the end of the sentence ' ... leading the way [13,31]' the punctuation mark is missing. 

- Figure 1: polysaccharide-based vectors is misspelled and is recommended to be rewritten correctly. 

- Line 254: '3.2. Methods of Formulation Viral Vectors' needs to be rewritten correctly.

- Line 493: 'through intracerebral and intracerebral injections [83]' does the sentence need to be modified or is correct as written? 

Author Response

Review 3:

Comments:

( ) I am not qualified to assess the quality of English in this paper
( ) English very difficult to understand/incomprehensible
( ) Extensive editing of English language required
(x) Moderate editing of English language required
( ) Minor editing of English language required
(  ) English language fine. No issues detected

Is the work a significant contribution to the field?

3/5

Is the work well organized and comprehensively described?

2/5

Is the work scientifically sound and not misleading?

3/5

Are there appropriate and adequate references to related and previous work?

4/5

Is the English used correct and readable?

2/5

Answer:

Our work has undergone major modifications. We tried to take into account all the comments of the Reviewer increasing the quality of our work. The manuscript has also been linguistically corrected again.

Comment:

“While I appreciate the effort and thought put into the this study, I believe it falls in a unique position that may not cater to the needs of either beginner or advanced researchers in the field. On one hand, the manuscript does not provide a sufficiently general and comprehensive overview suitable for beginners in the field. It lacks a broader context and fails to delve into fundamental concepts that would enable newcomers to grasp the subject matter more easily. As a result, novice researchers might struggle to fully understand the content without prior foundational knowledge. On the other hand, the manuscript does not offer the level of detail, depth, and novelty that would engage and benefit advanced researchers in the field. While it covers certain aspects reasonably well, it does not explore more intricate or cutting-edge areas that could contribute to the existing body of knowledge”.

Answer:

We would like to thank the Reviewer for taking the time to provide this opinion. However, we believe that the manuscript, due to its versatility, will attract a large audience of both new researchers entering the field and experienced research teams, who will be able to read the entirety of previous reports on the topics discussed. However, in order to improve the manuscript, we have included an additional paragraph on the limitations, industrial production and regulation associated with the use of viral vectors in therapy, which will be particularly useful for researchers starting out in this field of science. For more experienced researchers, we decided to prepare a table with patent registrations to highlight current research trends in the use of viral vectors in therapy as inspiration for future research.

We hope that, in its present form, our manuscript will be of value to its readers.

Comment:

“While the manuscript presents valuable insights, there are significant issues related to consistency, readability, and suitability of the English writing that need to be addressed”.

Answer:

We would like to thank the Reviewer for his fair comment on the linguistic correctness of the manuscript. Manuscript was revised extensively once more. All errors and typos have been corrected. Additional linguistic proofreading is also one of the steps in the procedure for publishing papers in MDPI, and therefore, in addition to our correction, there will be a linguistic evaluation at the publisher itself before the publication is made publicly available.

Comment:

“In relation to Figure 4, it is advised that the authors make adjustments to enhance reader comprehension. Initially, the left-side cell membrane implies a misconception regarding virus particle exocytosis and cell release, which is not accurate. By enhancing the figure, the process of virus particle endocytosis can be presented in a more lucid manner.”

Answer:

Thank you for your rightful comment regarding the clarity of the Figure 4. As suggested, Figure 4 has been extensively modified to better understand the mechanism.

Comment:

“Regarding the language and grammar, it is recommended to the authors to thoroughly proofread the article to correct grammatical errors, improve sentence structure, and ensure proper word usage. Few examples of language and grammatical errors are provided below: 

- Line 89: the sentence 'Another promising concept are also ...' contains potential grammatical error that needs to be corrected. 

- Line 137: at the end of the sentence ' ... leading the way [13,31]' the punctuation mark is missing. 

- Figure 1: polysaccharide-based vectors is misspelled and is recommended to be rewritten correctly. 

- Line 254: '3.2. Methods of Formulation Viral Vectors' needs to be rewritten correctly.

- Line 493: 'through intracerebral and intracerebral injections [83]' does the sentence need to be modified or is correct as written?”.

Answer:

Once again, we apologize for the language errors and thank the Reviewer for pointing them out. Accordingly, we have corrected the mentioned linguistic errors and corrected Figure 1. We hope that the text will be clearer and correct now.
